# Inferior Vena Cava Filter in Cancer-Associated Thrombosis: A *Vade Mecum* for the Treating Physicians: A Narrative Review

**DOI:** 10.3390/biomedicines12102230

**Published:** 2024-09-30

**Authors:** Agnese Maria Fioretti, Daniele La Forgia, Pietro Scicchitano, Natale Daniele Brunetti, Riccardo Inchingolo, Carlo Gabriele Tocchetti, Stefano Oliva

**Affiliations:** 1Cardio-Oncology Unit, IRCCS Istituto Tumori, “Giovanni Paolo II”, 70124 Bari, Italy; a.fioretti@oncologico.bari.it (A.M.F.); s.oliva@oncologico.bari.it (S.O.); 2Breast Radiology Department, IRCCS Istituto Tumori, “Giovanni Paolo II”, 70124 Bari, Italy; d.laforgia@oncologico.bari.it; 3Cardiology-Intensive Care Unit, Ospedale della Murgia “Fabio Perinei”, Altamura, 70022 Bari, Italy; 4Department of Medical and Surgical Sciences, University of Foggia, 71122 Foggia, Italy; nd.brunetti@unifg.it; 5Interventional Radiology Unit, “F. Miulli” Regional General Hospital, Acquaviva, 70021 Bari, Italy; r.inchingolo@miulli.it; 6Department of Translational Medical Sciences (DISMET), “Federico II” University of Naples, 80131 Napoli, Italy; cgtocchetti@gmail.com; 7Center for Basic and Clinical Immunology Research (CISI), “Federico II” University of Naples, 80131 Napoli, Italy; 8Interdepartmental Center of Clinical and Translational Sciences (CIRCET), “Federico II” University of Naples, 80131 Napoli, Italy; 9Interdepartmental Hypertension Research Center (CIRIAPA), “Federico II” University of Naples, 80131 Napoli, Italy

**Keywords:** venous thromboembolism, inferior vena cava filter, cancer

## Abstract

Cancer is a remarkable prothrombotic disease, and cancer-associated thrombosis acts as a dreadful omen for poor prognosis. The cornerstone of venous thromboembolism therapy is anticoagulation; however, in patients with venous thromboembolism who are not suitable for anticoagulation (contraindication, failure, or complication), the inferior vena cava filter appears a valuable option in the therapeutic arsenal. The recently heightened trend of steady rise in filter placement mirrors the spread of retrievable devices, together with improvements in physicians’ insertion ability, medico-legal issue, and novel and fewer thrombogenic materials. Nevertheless, the exact role of the inferior vena cava filter in cancer has yet to be endorsed due to a dearth of robust evidence. Indeed, data that support the inferior vena cava filter are weak and even controversial, resulting in discrepancies in the interpretation and application of guidelines in daily practice. In this narrative review, we aim at clarifying the state of the art on inferior vena cava filter use in malignancies. Furthermore, we provide a feasible, conclusive 4-step algorithm for the treating physicians in order to offer a practical strategy to successfully employ the inferior vena cava filter as a priceless device in the current armamentarium against cancer.

## 1. Introduction

Thromboembolism is the second leading cause of death in cancer after cancer itself; among all patients with venous thromboembolism (VTE), those with cancer account for 20%, encompassing deep vein thrombosis (DVT) and pulmonary embolism (PE) [1]. Notably, VTE is strongly correlated with increased inpatient mortality in malignancy due to a hypercoagulable milieu [1]. The risk of developing cancer-associated thrombosis (CAT) is significantly determined by four main interconnected factors: biomarkers and risk factors related to cancer, the patient, and anticancer agents [2]. Although the first-line CAT treatment is anticoagulation, a portion of patients do not fulfill its indication because of contraindication, failure, or complications of anticoagulation. In these insidious scenarios, inferior vena cava filters (IVCF) could be adopted in lieu of anticoagulants. They mechanically trap emboli from lower limbs/pelvis and prevent them from reaching the pulmonary arteries (PA) by catching the embolizing thrombi among metal struts [3]. Medicare A, claiming data from 1999 to 2008, stated that IVCF use frequency has doubled [4]. Nevertheless, to date, unanimous consensus is missing given the lack of strong evidence exploring mechanical VTE treatment with IVCF; on the contrary, anticoagulation is supported by research conducted with methodologic rigor [5]. The latest availability of retrievable devices has expanded their use to prophylactic intent but in the absence of proven efficacy; on the other hand, the increased filter placement complications give rise to unanimous concern [6]. Off-label IVCF implantation, clearly out of United States Food and Drug Administration (USFDA) indications, accounts for about 50% [7]. In the U.S., annually, 30,000–40,000 filters are deployed in cancer (30% of total placements) [8]. The purpose of this narrative review is to critically summarize the available updated evidence on IVCF efficacy and safety in cancer, focusing on the following items: indications, procedure, complications, retrievable filters, cancer population, and guidelines. A conclusive algorithm is offered for the treating physicians to manage IVCF in patients with CAT complications from a pragmatic viewpoint.

## 2. Methods

The database consulted for this narrative review was PUBMED (https://pubmed.ncbi.nlm.nih.gov/?otool=iitbisamlib, (accessed on 1 July 2024)). The main key words adopted were as follows: “vein or venous”, “thrombosis or clot or thrombus”, filter, inferior vena cava, and “cancer or tumor or malignancy”. All terms were combined with each other in order to obtain the maximum selection about this argument. One physician analyzed all the studies from PUBMED, combining each term in order to reach a complete overview on the literature background. All the included references are published journal articles written in English. Articles written in languages other than English and those on children were excluded.

## 3. Indications to Inferior Vena Cava Filter Placement

Indications to place a filter could be categorized into absolute, relative and prophylactic (Table 1).

Patients with acute VTE and contraindications, those whose anticoagulant therapy fails, and those who develop complications are candidates for mechanical VTE treatment with IVCF [9]. The International Society on Thrombosis and Hemostasis guidance recommends in cancer patients the same indications as in those without cancer. They do recommend against IVCF insertion when contraindications to anticoagulation are absent and in favor of resuming anticoagulation at filter removal time when contraindications have been resolved. In cancer patients with acute VTE and thrombocytopenia < 50 × 10^9^ L^−1^, if platelet transfusion in not advisable, the authors suggest a retrievable IVCF insertion to be removed once the platelet count recovers [10]. After the advent of new-generation devices with retrievable intent, filter use increased, transcending guidelines-based recommendations and widening indications to new categories of patients: free-floating large proximal/iliocaval DVT, poor compliance to manage therapeutical anticoagulation, high risk to be anticoagulated, massive PE/iliocaval VTE treated with thrombolysis/thrombectomy in patients at risk for further PE regardless of anticoagulation status, chronic PE treated with thromboendoarterectomy, VTE with limited cardiopulmonary reserve, and filter failure. In addition, research addressing relative indications has been undertaken mostly in selected populations including trauma patients, those undergoing bariatric surgery, and pregnant and pediatric patients but not in those with malignancy [6]. Patients without VTE who need to prevent it due to a high VTE risk and cannot receive anticoagulant prophylaxis could be suggested IVCF with a temporary intent. including high-risk surgery/invasive interventions, medical prothrombotic disease. and high-VTE-risk trauma patients. Nevertheless, currently, prophylactic indication is not fully driven by evidence [6].

## 4. Inferior Vena Cava Placement Procedure

A preoperative cavography should be performed to study anatomy, size, patency, and location of the renal veins and to detect the possible IVC anatomic variants. Depending upon device design and delivery sheath diameter, most filters are placed by accessing the femoral or jugular veins, while less common alternatives include the antecubital and subclavian veins [11]. The procedure is often performed using fluoroscopy in endovascular angiogram suites or at bedside if intravascular ultrasound-guided imaging is available in unstable or immovable patients [12]. The usual target-landing zone for ICVF is the infrarenal IVC at L3 vertebral body level caudal to renal veins. The IVC diameter at the level of the target-landing zone is crucial since each filter is appraised for a maximum IVC diameter, above which the risk of filter embolization is increased [13]. The delivery sheath is placed at the desired final site of the filter, where the device must be moved up to the end of the delivery sheath, and the delivery sheath is then withdrawn. Filter release follows the manufacturer’s instructions. Cavography is repeated, using the same parameters, to check the final filter location at the end of the procedure (Figure 1).

## 5. Inferior Vena Cava Filter Placement Complications

Although IVCF placement is a generally safe technique with a low major complication rate (roughly < 0.5%), clinical concern is not negligible and relevant, mostly, when the filter is placed for prophylactic ICV interruption. The complication profile is time-dependent; most occur around the time of the IVCF insertion (≤30 days following placement), while most of them are related to long-term use (also months following insertion), with heightened frequency after the first 30 days since placement (Table 2).

Accordingly, the ICVF should be removed as soon as possible once PE risk is elapsed [14]. Deso and coworkers identified 23 IVC filter types in order to find device-specific risk for a prompt decision between follow-up and retrieval. The IVCF geometries were conical, conical with cylindrical element, biconical with cylindrical element, conical with umbrella, helical, complex, and spiral with umbrella. Conical filters accounted for the highest reported risk of perforation (90–100%), whilst filters with cylindrical or umbrella elements were associated with the highest reported risk of IVC thrombosis (30–50%) [15].

### 5.1. Delayed Filter Removal

Although the manufacturer’s published instructions for use, a review of 37 trials including 11 prospective clinical trials (6834 patients) revealed that filter removal rate was dramatically just 34%. Indeed, a database on patients with filters was missing, and at insertion time, patients often did not receive a scheduled time date for removal and, as a result, were lost to follow-up. Along with a low removal rate, the PE rate was very low (1.7%), highlighting the beneficial effect of a retrievable filter in preventing VTE events [14]. Moreover, before any filter removal, a cavogram should be performed to rule out the presence of a thrombus. Therefore, effort should be made to remove the filter after the requested reason subsides [16]. Nonetheless, retrievable filters should remain in situ on a permanent intent in case of removal failure or its continued need. In a single-center, retrospective assessment of 484 retrievable IVCFs (10.5% of patients had cancer), more than half (52.7%) of the devices were converted to permanent filters based on the clinical decision of the treating clinician. Regardless of a strict tracking system by mail and phone, nearly half of patients were lost to follow-up. Roughly 24% of patients refused filter retrieval based on the inexact advice of another physician. Most patients with filters used with a permanent intent belonged to certain categories, including older patients and those with comorbidities. To overcome this challenge in management, the authors suggest performing filter retrievals during hospital stay at discharge [17]. To confirm these findings, Ribas and colleagues retrospectively assessed 271 patients inserted with IVCF (permanent in 24.4% of patients and temporary in 75.6%) in 83% cases due to contraindication to anticoagulation for acute VTE; 66% of filters inserted with temporary intention were retrieved. Filters were not retrieved mainly for lack of planning/follow-up (57.9%); in addition, cancer was the leading cause of death at 10-year follow-up (49%). Patients who had a filter retrieved were younger, mostly without cancer, and were anticoagulated after placement insertion. Extended anticoagulation was not correlated with survival benefit [18].

### 5.2. Caval Wall Penetration

Filters slightly penetrate the vena cava wall for strong anchoring in order to avoid migration and distant embolization to the heart. However, it becomes a caval wall perforation when it reaches > 3 mm beyond IVC lumen or within an adjacent structure (small bowel, pancreas, ureter, or aorta) with one or more filter struts, mostly in long-term filter dwelling. Incidence of symptomatic penetrations is reported of 0.3%, mainly 3 weeks after deployment. Not-specific symptoms (mainly abdominal pain) or retroperitoneal bleedings are very uncommon and rare but require immediate removal, necessitating more aggressive and inquisitive techniques. Asymptomatic caval perforation’s radiologically and incidentally detected rates are noted in up to 40% of patients, and its management is less well established [19]. However, if penetration involves adjacent structures, removal should be undertaken to avoid further progression, or in case the filter is not removed, a CT scan follow-up should be applied. Medline database was searched, and 9002 patients inserted with an IVCF were analyzed, resulting in an occurrence of filter penetration of 19% (symptomatic patients accounted for 1/10 of all penetrations) with adjacent organ/structure involvement of 19%. Of all penetrations, 5% were considered major complications requiring surgery or endovascular retrieval. Unplanned interventions with or without surgery were required in 4.5% of cases [20].

### 5.3. Filter Fracture

A large retrospective analysis of 548 patients inserted with IVCF showed arms and legs fractures of only 6%, with removal success of the entire body or of the components (arms and legs) of 98.4% and 53.4%, respectively. Distal embolization, mainly to PAs, of the fractured filter components was only 13% without immediate overt sequelae. Filter fracture increased in longer dwell times; accordingly, fractured filters can be successfully removed without severe clinical complications [21].

### 5.4. Filter Migration and Embolization

Filter migration (a change in filter position > 2 cm on imaging from initial placement location) is a very rare complication, mainly asymptomatic and diagnosed by abdominal CT scan, due to metal fatigue. However, if the filter crosses the ostium of a large draining vein, a visceral venous thrombosis could occur. One or more struts or rarely the whole filter can migrate and cause embolization. The broken filters must be promptly extracted, and generally, they are entrapped in the right atrium [14]. A single-center, retrospective cross-sectional study reported 16% of 80 patients inserted with IVCF had at least one strut fracture, and 25% had fragments embolization. Of note, 3 out of 28 patients had with life-threatening cardiac complications due to fragments migration to the heart. Patients with VTE and the medical community should be educated about the possible complications subsequent to IVCF deployment. Indeed, shortness of breath, pleuritic chest pain, dizziness, and syncope could be presumed to be symptoms related to a recurrent VTE event, in the absence of a suspicion of right-heart perforation and tamponade, due to filter fracture and migration. A prompt recognition of complaints by filter recipients could enable an immediate alert for physicians and prompt removal [22].

### 5.5. Post Thrombotic Syndrome

Patients who use IVC filters as secondary DVT prophylaxis, compared to patients using anticoagulants, have an increased risk of DVT complicated by post-thrombotic syndrome (PTS). PTS is a chronic condition, sometimes with persistent symptoms such as lower-limb swelling, paresthesia, pain and skin eczema, hyperpigmentation, and venous ulcerations. PTS reduces the quality of life of patients and requires expensive and long-term treatments [14]. Pharmacological anticoagulation of VTE reduces symptoms, VTE recurrence, and propagation but does not lyse the clot, therefore enabling PTS development that occurs in up to 50% of patients with proximal DVT within 2 years. An increase of PTS in filter recipients is still uncertain, although current data suggest an increased PTS risk in long-term dwelling of permanent IVCF [23].

### 5.6. IVCF Thrombosis or Occlusion

The occurrence of chronic thrombosis is not a negligible event after filter insertion. It is mostly asymptomatic and leads to formation of tributaries that bypass conduits and lastly hesitate in PE. Moreover, ICVF thrombosis can complicate its removal with several side effects, including decreased protection against PE, filter migration, PTS, and venous stasis. However, modern filter materials contribute to avoiding thrombosis and stenosis due to a less thrombogenic surface compared to older ones [14]. Moreover, as shown in a retrospective investigation that involved 440 patients, the rate of filter thrombosis decreases after longer dwell time, as detected by preretrieval cavogram, suggesting an additional anticoagulant period if thrombus is found to foster later retrieval attempts [24]. Among 1718 patients (64 had cancer) with IVCF, 18.6% developed device thrombosis (18.6% of subjects, >98% asymptomatic) with rare progression (1.7% of patients) to IVC total occlusion, which was mainly asymptomatic. Regression of filter thrombus on anticoagulation was low (51.2%); thus, it could be avoided in high-bleeding-risk patients. Furthermore, anticoagulation had little influence on PE occurrence in patients with filter thrombus; therefore, the positioning of IVCF in anticoagulated patients is still a dilemma [25].

## 6. Retrievable Vena Cava Filters

There are two basic classes of IVC filtration devices available to date: permanent and retrievable. Permanent IVCF, used since the 1970s, is indicated in patients with long-term contraindications to anticoagulation or long-term required TVE prophylaxis. Retrievable IVCF, which have been in use since the late 1990s, can be left in position or retrieved when the risk of PE or the contraindication to anticoagulated has been resolved, acting as a “bridge” to anticoagulation [26]. Retrievable filters include two types: temporary filters and optional filters. Temporary filters are tethered to a wire, and the removal timeframe is device-specific to avoid its incorporation into the IVC wall [27]. Optional filters appear as a more attractive device since they are designed to be removed at a planned time and directly become incorporated in the IVC wall by a hook [28]. Imberti and collaborators tested retrievable IVCF in 30 patients (4 with cancer) with acute VTE and contraindication to anticoagulation (86%), primary prophylaxis after major trauma (7%), and surgical prophylaxis in very-high-VTE-risk patients (7%). Although it was a limited sample size, the authors observed a 10% rate of trapped emboli in the filter after an 18.2-month follow-up, 3% rate of asymptomatic migration to the heart, and 7% rate of DVT recurrences. Importantly, the study suggested that filter retrieval after 3 months could be unsuccessful: 100% of filters were removed with success, while when the retrieval attempt was performed beyond 3 moths from placement, it was possible in only 50% of cases [29]. An audit on the IVCF in the United Kingdom from three centers over 12 years reported that of a total of 516 filters placed, the retrieval rate was only 32%. The long-term in situ placement could be due to manufacture design (intent at placement), necessity (difficult retrieval), or omission (loss to follow-up) [30]. Kim and coworkers investigated the outcomes of 702 recipients (60.8% placed with retrievable IVCF and 39.2% with permanent IVCF) for 5 years in a retrospective cohort study, resulting in a similar incidence of subsequent VTE and filter-related complications. The new symptomatic PE rate was 4% and 4.7%, symptomatic IVC thrombosis was 1.1% and 0.5%, and new or worsening DVT was 11.3% and 12.6%, respectively, in the permanent filter group and in the retrievable filter group. The first group mainly comprised cancer patients and older patients, whereby the second group included younger patients with a filter due to a large proximal thrombus thrombolysis. The paradigmatic discovery was that, although the retrievable filters were supposed to be placed for a temporary reason, only 10.8% were successfully retrieved because of the persistence of contraindications to anticoagulation or the presence of a thrombus in the filter or close to it [31]. About 50% of retrievable filters are never removed, regardless of indications. A reasonable removal within approximately 12 weeks from insertion avoids long-term DVT sequelae [32]. A reminder-based strategy to follow-up IVCF recipients was successful in reducing patients lost at follow-up, as reported in a retrospective analysis on the management of 1127 filters. A structured follow-up consisting of periodic mail sent to the patients at a periodic interval once removal time was reached resulted in a statistically significant gain in timely filter removal (58.4% vs. 24% without follow-up method, *p* < 0.001). Radiologists used a “one-device-fits-all” strategy since they also placed retrievable filters in patients with permanent indication. Of note, only 16.5% of patients were lost at follow-up [33]. Taking into account the exponential rise in IVCF placement, mostly due to new-generation retrievable filters, followed by mounting evidence of reported adverse side effects, the USFDA issued a safety warning (9 August 2010) recommending that *“implanting physicians and clinicians responsible for the ongoing care of patients with retrievable IVC filters consider removing the filter as soon as protection from PE is no longer needed*” [34]. This alert was updated (6 May 2014) with a subsequent safety communication, adding that the “*FDA encourages all physicians involved in the treatment and follow-up of patients receiving IVC filters to consider the risks and benefits of filter removal for each patient. A patient should be referred for IVC filter removal when the risk/benefit profile favors removal and the procedure is feasible given the patient’s health status*” [35]. Moreover, the USFDA, in accordance with a decision analysis based on a mathematical model, suggested that if filter indications subsides, the benefit/risk profile favors filter removal between 29 and 54 days after placement [32]. A consequential decline in filter placement in the U.S. was reported in a retrospective analysis from 2010 and 2019 that investigated 823717 IVCF insertions (78.3% for VTE treatment and 21.7% with prophylactic intent), with an aggregate rate reduction of −8.4% [36]. Uberoi and collaborators published the experience of the Interventional Radiology IVCF Registry, evaluating 1434 IVC filters. Retrievable filters were used in 73% of patients, even when the indication was for permanent placement. Patients with underlying comorbidities and poor prognosis were more frequently planned for a permanent filter placement to reduce the amount of hospital access and interventional procedures. The in-hospital mortality rate was 8.1% after filter placement, mirroring that the subset of patients undergoing filter placement that belonged to the high-risk population, whose death could probably be due to the severe underlying clinical diseases. Complications were similar between permanent and retrievable filters and were related to the training of the operators, being higher for a limited learning curve and experience of the implanting physician (<25 IVCF insertions). Retrieval success was higher if attempted <9 weeks after placement; nevertheless, 40% of registry patients were lost at follow-up. Thus, the authors suggested that at the time of placement, if the intent is retrieval, the interventional radiologist should already plan retrieval at a suitable interval to avoid long-term filter-related complications [37]. The MAUDE (USFDA Manufacturer and User Facility Device Experience) database compared the adverse IVCF-related events in 1057 recipients, revealing a higher frequency (86.8%) among those with retrievable filters (75% of all filters) compared to those with permanent filters (13.2%) (*p* < 0.0001). There was a significantly higher frequency of each adverse event (fracture, migration, limb embolization, tilt, IVC penetration, DVT/PE, IVC thrombosis, and malfunctions) in the retrievable group compared to the permanent group. Fracture was the most frequent adverse event among retrievable filters, while placement malfunctions were the most frequent complication among permanent filters [38]. The CIRSE (Cardiovascular and Interventional Radiological Society of Europe) published a registry of 671 retrievable filters, revealing a retrieval rate of 92% (mean dwell time: 90 days) and a retrieval failure of 8% (mean dwell time: 145 days), with a low complications rate (major 0.03% and minor 2%). Indications to IVCF insertion were absolute in 40% of cases, relative in 31%, prophylactic in 24%, and missing in 5% [39]. A systematic review of retrievable IVCF efficacy and safety in >1000 patients showed neither an increase in DVT risk nor a significant reduction in PE or PE-related mortality [40]. The. main standards for filter retrieval include resumed possibility for anticoagulation if still needed, retrieval safety, life expectancy > 6 months, and patient’s preference. A Swiss tertiary-care university hospital analysis, over a 19-year period in 920 patients (28.8% had cancer), established that 40.65% of retrievable filters were left in place due to several reasons: lack of follow-up (22.34%), prolonged contraindication to anticoagulation (20.51%), technical issues (17.40%), and short life expectancy (17.22%). The most frequent IVCF-related complication was filter thrombosis. Most of the patients had primary indication to a retrievable filter [41]. Filter retrieval failure varies from 0 to 20% due to the following reasons: filter embedded in IVC wall (4.7–6.2%), filter angulation > 15° (1.9–16%), severe leg penetration > 15 mm (0.3–2.7%), and prolonged dwell time (10–29%), mostly related to lack of follow-up. For this reason, Rezaei-Kalantari and coworkers aimed to optimize clinical filter management, proposing an IVCF workflow strategy starting from filter insertion to medical report, focusing on a planned clinical visit with the implanting physician 5 months after placement [42] (Table 3).

## 7. Inferior Vena Cava Filters Outcomes in Cancer

Cancer develops a prothrombotic state per se due to hypercoagulability, and in patients placed with an IVCF, the clot burden is primarily driven by treatment-related factors [43] (Figure 2).

### 7.1. Evidence among Medical Cancer Patients

The only prospective randomized trial to date, the landmark Prepic study first published in 1998 [44] and then updated in 2005 [45], compared patients with acute proximal DVT with or without PE treated with anticoagulants alone for at least 3 months versus those with “add on”, non-retrievable IVCF. Of the 400 enrolled patients, cancer patients comprised 16% of the filter group and 12% of the nonfilter group. At 12 days after randomization, filters had no impact on mortality, but a statistically significant reduction of 78% in PE risk was found in the inserted patients (odds ratio OR 0.22, 95% confidence interval CI 0.05–0.09, *p* = 0.3), but this initial protecting benefit was not confirmed at 2 years [44]. At 8 years, the findings showed a lower rate of PE in filter recipients compared to those without (6.2% vs. 15.1%, *p* = 0.008), although it was not statistically significant, at the cost of a higher DVT rate, mainly of the lower limbs, due to filter occlusion (20.8% vs. 11.6%; OR 1.87; 95% CI, 1.10–3.20; *p* = 0.02). However, roughly half of the patients stopped anticoagulant therapy after 6 months of treatment, and the majority of recurrences occurred after anticoagulation withdrawal. No difference was found in long-term mortality and PTS between groups; therefore, filter placement seemed not to confer survival benefit [45]. The trial established an increased risk of bleeding in the group with anticoagulants without filter (OR 1.49, CI 0.53–4.20, *p* = 0.44). The authors balanced the risk–benefit profile of IVCF in patients with VTE, suggesting that the protection offered against PE is counterbalanced by an enhanced DVT risk; thus, filter use should not be systematic but could be beneficial in selected patients with high-risk conditions, such as PE as the first event, idiopathic VTE, and those with cancer. The Prepic trial did not assess the filter’s role as a prophylactic device in patients without VTE. Overall, these considerations together with the meaningful underrepresentation of cancer patients in the Prepic study underscore that until now, the use of IVCF in cancer has not been clearly tested, and further research is drastically needed. Notably, the exclusion criteria encompassed contraindication to or failure of anticoagulant therapy, which are the absolute indications to IVCF placement, and 94% of patients received at least 3 months of anticoagulation; accordingly, the trial did not offer answers on the outcomes of patients with the main current indications to IVCF placement. Of note, there is no clinical trial so far designed to compare IVCF to anticoagulation for VTE treatment. In the follow-up study Prepic2, 399 patients hospitalized for severe acute concurrent PE and lower-limb DVT plus at least one high-risk criterion for recurrence (age > 75, active cancer, or cardiac or respiratory failure) were anticoagulated for 6 months and randomized to receive or not retrievable IVCF; in the filter group, history of cancer was present in 7.5% of patients and 11.6% of the control group. There was no significant bleeding difference between groups. At 3-month follow-up, symptomatic recurrent PE was not reduced in the filter group; furthermore, there was no advantage in term of mortality. There was not an increase in DVT recurrence in inserted patients, likely according to study protocol requesting IVCF retrieval after 3 months. Findings were similar at 6 months, with support for avoiding IVCF placement in this subpopulation of patients who can be anticoagulated. As in the Prepic study, patients with contraindication to anticoagulation were excluded from the study [46]. Barginear and colleagues retrospectively examined 206 VTE cancer patients with localized oncological disease placed with IVCF and randomized to anticoagulation, anticoagulation and filter, or filter alone. The recurrent PE rate was 89% in IVCF patients versus 11% in those on anticoagulation alone; patients inserted with IVCF were 1.9 times more at risk of death compared to those on anticoagulation alone (Hazard Ratio HR 0.528; 95% CI = 0.374 to 0.745). These findings suggest the absence of benefit in IVCF use in cancer patients for survival, considering cancer patients placed with IVCF already have a dismal prognosis [47]. The patients from the Nationwide Inpatient Sample who were discharged from short-stay hospitals in the U.S. from 1998 to 2009 were selected based on PE, solid cancer, age more than 30 years old, and stable clinical conditions. Among these patients, the filter recipients showed a case fatality rate (10.2%) lower than those without a filter (14.9%). In the subset of patients with hematological malignancies, the benefit resulted only for the elderly [48]. A retrospective observational study was undertaken to focus on the use of IVCF in cancer patients with VTE about the higher frequency of IVCF placement in cancer patients compared to noncancer patients. Indeed, an IVCF was placed in 19.6% of cancer patients, but the placement frequency varied markedly among California hospitals and cancer types, ranging from 0% to 52% [49]. Additionally, the same authors stated that IVCFs were more likely to be inserted in urban, larger, and private hospitals, probably due to the larger availability of radiologists or vascular surgeons skilled in placing IVCF. These social considerations could be crucial in cancer patients who need more frequent hospital access, implying a worse adherence to anticancer treatments. IVCF was more frequently inserted in some cancer sites, such as the brain, melanoma, pancreas, female genital tract, colon, and urinary tract, possibly due to a perceived high bleeding risk. Indeed, brain cancer was associated with over 4-fold increased odds of IVC placement, with a higher risk of intracranial hemorrhage while on anticoagulation. Importantly, it was reported that overall, only 21% of cancer patients with an IVCF inserted had a strict contraindication to anticoagulated (surgery or bleeding). Interestingly, 62% of patients with IVCF developed complications [49]. A small sample size population of VTE cancer patients (64) was prospectively explored in 2016 for clinical outcomes, resulting in no reduction in short-term mortality or in PE recurrence (within 180 days); instead, there was both a 60% increased DVT recurrence risk and a 20% risk of subsequent bleeding. The majority of patients did not have a contraindication to anticoagulation. Accordingly, the authors suggested avoiding a routine use of IVCF since it caused more harm than benefit [50]. A population-based study analyzed hospital discharge records of 14,000 cancer patients who were admitted to a California hospital for acute DVT or PE. Approximately 20% of patients were placed with an IVCF irrespective of clear evidence (only 21% indications-based). Outcomes showed neither 30-day survival benefit nor a reduction in 6-month PE incidence, while on the other hand, patients with an IVCF had a 2-fold increase in DVT recurrence [51]. Leiderman and colleagues retrospectively analyzed 147 cancer patients, comparing those who required an IVCF with those who could be anticoagulated. The risk of death was 8.83 higher in the filter group (64% had metastases) compared to the anticoagulant group (32% had metastases), highlighting how filter placement is a marker of advanced disease and worse prognosis. Notably, in the filter group, the risk of death was 2.24-fold higher if medically indicated as compared to patients who had a surgical indication to insertion, possibly due to the better clinical conditions and performance status of patients facing surgery [52]. In a population-based cohort study of 88,585 cancer patients with acute lower-limb DVT, in 38.1% of cases, an IVCF was placed mainly due to gastrointestinal bleeding, intracranial hemorrhage, and coagulopathy. The PE-free survival was increased in the patients with filter compared to patients without, even after accounting for death as a competing risk and comorbidities. Filter placement did not increase DVT recurrence after accounting for use of anticoagulants and bleeding risk factors [53]. According to the data from the RIETE, an ongoing international multicenter registry of 17,005 cancer patients with VTE, 270 underwent IVCF placement due to absolute or relative contraindications to anticoagulation. After 1-month follow-up, recipients showed a significant lower risk of PE-related mortality at the cost of an increased symptomatic VTE recurrences, without significant change in major bleeding risk or rate of all-cause mortality. These data from a non-selected population confirm the possible benefits related to IVCF also in the real world for patients with cancer [54]. A large cohort study (25,788) evaluated cancer patients with VTE. Less than 10% of patients were placed with IVCF, and moreover, the retrieval rate was low (7.5%), although one-third of patients were tolerant towards anticoagulation 30 days after placement. In addition, filter use (placement/retrieval) was associated with nonclinical reasons including income and region since patients with the highest incomes were more likely to be implanted with filters and the filters to be more often retrieved. These data underscore possible patient disparities due to sociodemographic conditions, discriminating vulnerable cancer patient populations, including those with lower health care literacy, health care access, and financial resources [55]. A multicenter, prospective, open-label, nonrandomized investigation evaluating 1421 (37.3% had active cancer) IVCF (44.5% retrievable filters)-implanted patients (81.6% due to contraindication/failure to anticoagulation; 8.9% were prophylactic) were closely monitored at 3, 6, 12, 18, and 24 months after implant with in-person visits and telephone calls. The 12-month outcomes showed a VTE event incidence of 6.5% within the first year after placement, uncommon and minor procedural adverse effects, and only 0.5% filter-related adverse effects. The retrieval rate was high (49.3%), suggesting that if the implanting physician enacts a placement plan for if and when the filter will be removed, with frequent re-evaluations, the amount of patients lost at follow-up will be reduced, and the indicated filter removal will be safe and prompt. This proposed strategy could be crucially favorable, especially in cancer patients, since those implanted with IVCF have poor general clinical conditions [56].

### 7.2. Evidence among Medical Cancer Patients with Advanced Disease

Jarrett and coworkers reported, in 116 late-stage cancer patients (78.4% had stage IV disease; 42 died of cancer within 6 weeks) with VTE carrying IVCF, a very poor survival rate: 68.8% at 30 days, 49.9% at 90 days, and 26.8% at 365 days, arguing no benefit in IVCF placement in a subpopulation of cancer patients with such a dismal prognosis. Importantly, 1-year survival for stage IV disease was only 13.7% versus 77.9% for stage I–II–III patients, with a statistically significant difference (*p* < 0.001). The authors suggested no treatment in advanced-stage cancer patients with VTE and contraindications to anticoagulation and in those with recurrent TEV despite anticoagulation; they suggested the same anticoagulant dosage or a more intensive therapy instead of filter placement [57]. Similar conclusions came from a study conducted in 24 brain cancer patients with lower-limb DVT on anticoagulation randomized to filter vs. no filter placement. Indeed, IVCF was effective in reducing PE rate but was not cost-effective. It could be more reasonable to insert an IVCF in cancer patients with longer life expectancies [58]. Other authors retrospectively reported on 308 cancer patients (80.5% had locally advanced or metastatic disease) with VTE, highlighting a higher survival rate compared to those reported by Jarrett and colleagues (81% at 30 days, 60% at 90 days, and 35% at 365 days); nevertheless, the absolute number was higher, and the percentage of patients with advanced disease was lower (58.2% vs. 78.4%). IVCFs were effective in PE-related death prevention without higher filter placement complications. IVCF thrombosis was reported in 4.5% of patients, PE in 1.3%, and retroperitoneal hemorrhage in 0.7%. However, patients with advanced disease had the lowest survival rate [59]. The objective of a case–control study was to retrospectively determine the outcomes in 55 consecutive stage III and IV cancer patients with VTE who were inserted with IVCF, as identified in a tertiary-care facility. The filter prevented PE in 94.5% of patients, while complications occurred in 7.3% of patients; notably, 23.6% of advanced-stage cancer patients survived less than 30 days following filter insertion. Therefore, there was no survival benefit in the inserted group compared with the group of not-inserted patients, and survival was mainly driven by the malignant burden. In this particular subset of cancer patients, a low-level anticoagulation or no treatment could be the best management options [60]. Another trial was prospectively conducted in 64 cancer subjects mostly in IV stage disease and 2/3 Eastern Cooperative Oncology Group (ECOG) performance status who were randomized to anticoagulation with fondaparinux alone or fondaparinux with IVCF placement. Adding an IVCF to fondaparinux did not show a benefit in terms of safety (major bleeding complications were only <5% and similar in both arms), efficacy (complete VTE resolution occurred in 51% of patients within 8 weeks), or survival (a negative trend in IVCF arm was due to a higher rate of advanced colon and pancreatic cancer with dismal prognosis assigned to IVCF arm). The complication rate in IVCF was low (7%) [61]. This raises the question of the necessity of inserting IVCF in advanced-stage cancer patients since in this subpopulation, it seemed to offer limited benefits, not improve survival, and decrease their quality of life. Indeed, in a retrospective evaluation of 107 cancer patients (>90% had advanced-stage disease) with IVCF inserted, 67% survived less than 3 months after IVCF placement. In this study, indications to IVCF insertion were found in 16.8% of patients with anticoagulation failure and in 79.4%, there were contraindications to anticoagulation. On the other hand, the ICVF insertion procedure was not free of complications (recurrent DVT in 9.3% of patients and PE in 2.8%) [62]. Mahmood and colleagues looked at outcomes in 154 cancer patients with VTE, comparing those with metastatic disease with those with localized cancer; all had a contraindication to anticoagulation and were implanted with IVCF. The 30-day mortality was 16.5% among metastatic patients, while 6-month and 1-year survival was 55% and 37%, respectively, in the metastatic setting in comparison to 78% and 64%, respectively, in the nonmetastatic setting. Of note, the authors found that patients with a reduced life spam should be selected more often in filter placement since metastatic patients had more complications compared to nonmetastatic patients (25% vs. 11%). In addition, the retrieval rate was lower in metastatic patients compared to nonmetastatic (31% vs. 58%), and mostly, the time to resume anticoagulation was longer (5.5 vs. 2 days) [63]. ICVF placement is debatable in patients with advanced cancer since benefits are limited [64]. There is special concern for IVCF placement in this clinical setting due to a low rate of survival; therefore, risk–benefit assessment of filter placement remains challenging, and decision making should be done on a case-by-case basis [65].

### 7.3. Evidence among Surgical Cancer Patients

In 10 cancer patients (mostly with pelvic sites disease) with VTE already in anticoagulant therapy for whom surgery was contemplated, IVCFs were prophylactically implanted with a safe post-operative course [66]. A retrospective, small-sample analysis (38 patients) was undertaken to test the efficacy and safety of IVCF placement in gynecological malignancies, including uterine, cervical, and mostly ovarian cancer undergoing major surgery, for 6-month follow-up. No complications and no death were found, and among the 11 recipients who had the filter retrieved, in 82%, the procedure was uneventful. Filter retrieval was established based on cancer stage, chemotherapy, time since filter deployment, and life expectancy [67]. Matsuo and collaborators investigated the perioperative use of IVCF in 274 patients (52.6% with VTE) who underwent primary cytoreductive surgery for ovarian cancer, suggesting an increased risk of hematogenous metastases and decreased survival. The mechanisms supposed by the authors include the proinflammatory effect triggered by the IVCF as a foreign body and the platelets activation promoted by the IVCF [68]. A single-center, retrospective 10-year analysis of 250 cancer patients (77.2% had proximal lower-limb DVT) indicated that 18.4% received an IVCF (51.2% were retrievable) placement since undergoing surgery and could not be anticoagulated thereafter, and only two patients had filter removal. Patients who died early after filter placement were those with metastatic disease, with lower body max index, who were undernourished, and with ECOG performance status >3. Thus, before filter placement, it is useful to weigh patient prognosis to save resources and suffering [69]. On the other hand, the positioning of IVCF in cancer patients who undergo major surgical procedures seems a quite safe procedure with minimal filter-related complications, low rates in PE, and negligible DVT/bleeding risk [70]. As these patients are at higher risk for both recurrent thrombosis and bleeding events, IVCF placement might be needed [71].

## 8. Inferior Vena Cava Filter Indications: A “Head-to-Head” Major Guidelines Comparison

The CIRSE guidelines endorse IVCF in patients with PE or DVT and contraindication/failure/complication to anticoagulation. They also advocate relative recommendations, including severe PE with residual DVT at high risk of recurrent PE, free-floating iliocaval thrombus, severe cardiopulmonary disease and DVT, poor compliance with anticoagulants, severe trauma without documented VTE events, high-risk patients (prophylactic use in surgical patients and those immobilized), local fibrinolysis for DVT, and pregnant patients with DVT at cesarian section/birth [72]. The European Society of Cardiology recommends IVCF in patients with acute PE and absolute contraindication to anticoagulation or PE recurrence despite anticoagulant treatment and does not recommend its routine use [73]. The Society of Interventional Radiology (SIR) guidelines advocate IVCF placement in acute VTE patients with contraindication to anticoagulation, albeit they do recommend against filter placement in trauma patients and for anticoagulant failure and prophylactic purposes in patients undergoing major surgery [74]. The American College of Chest Physicians (CHEST) guidelines recommend in favor of IVCF in patients with acute proximal leg DVT and contraindication to anticoagulation, while they do recommend against IVCF in addition to anticoagulation in acute leg DVT [75]. The American Society of Hematology (ASH) guidelines confirm the indication to filter placement in VTE cancer patients with contraindication to anticoagulation, and they do recommend against “add-on” an IVCF in cancer-related recurrent VTE despite anticoagulation [76]. The International Initiative on Thrombosis and Cancer (ITAC) guidelines indicate placing IVCF in VTE cancer patients if anticoagulation is contraindicated or if VTE events recur despite optimal anticoagulation; they do recommend against IVCF use for routine prophylaxis [77]. The American Society of Clinical Oncology (ASCO) guidelines recommend VCF in acute (<4 weeks), life-threatening VTE events and absolute contraindication to anticoagulation and as an “add-on” to anticoagulation in patients with VTE recurrence or extension despite optimal anticoagulation. They endorse no indications for temporary contraindication to anticoagulation (e.g., surgery), chronic (>4 weeks) VTE, or primary prevention/prophylaxis [78]. The European Society of Medical Oncology (ESMO) guidelines indicate VCF in acute and life-threatening VTE with absolute contraindication to anticoagulation and as an “add-on” to anticoagulation in patients with VTE recurrence or extension despite optimal anticoagulation [79]. The British Society of Hematology guidelines recommend temporary IVCF placement in acute VTE and contraindications to anticoagulation or when its interruption is requested within the 4 weeks after acute VTE diagnosis. In addition, they recommended against routine IVCF use in cancer patients [80]. Overall, the main current guidelines overlap in advocating only one among the three absolute indications to IVCF placement, namely the contraindication to anticoagulation in patients with acute VTE, but they mostly differ on the relative and prophylactic recommendations. Low adherence to current guidelines is not negligible (37.8%) in clinical practice, especially in the fragile-patient categories: elderly, comorbid, and cancer patients. A particular concern refers to cancer patients who have the highest mortality once placed with IVCF, while at the same time, they could gain the highest benefit from filter placement [81]. Future directions could be addressed to establish clear guidelines with the aim to identify the perfect balance, especially in the prophylactic setting, between benefits and risks to IVCF placement [82].

## 9. How to Use Inferior Vena Cava Filter in Cancer: An Updated Suggested Algorithm

We propose, in accordance with current evidence documented so far, a practical strategic algorithm to use IVCF in cancer patients based on a 4-step approach. Step 1: Consider ICVF in active cancer patients with acute VTE events and absolute contraindication to anticoagulants. Step 2: At the date of IVCF placement, fix a scheduled time date for removal, considering the preferences of the patient, who will receive educational instructions on signs and symptoms of IVCF-related complications to alert nurses/treating physicians. Step 3: Check patients’ clinical condition using a tracking system by mail or phone to test removal feasibility and verify possible IVCF complications for immediate interventions. Step 4: Remove the IVCF at 29–54 days from placement at the date already planned with the patient (Figure 3).

A multidisciplinary team (hemostasis and thrombosis experts, interventional radiologists, and vascular surgeons) could facilitate the application of this algorithm in clinical practice [83].

## 10. Conclusions

Cancer is the cause of a hypercoagulable state, increasing the risk of developing VTE 4- to 7-fold compared to noncancer patients. Anticoagulation is the standard of care for patients with CAT, but it is not suitable for those with contraindications to anticoagulation or in scenarios of ineffective anticoagulation or complications with anticoagulation. In these challenging scenarios, the IVCF could be a viable alternative, taking into account that its placement is a low-risk procedure. The concept of IVC interruption by filters is still evolving, with exponentially broadened indications, noticeably driven by the commercial availability of retrievable devices, that are not supported by good-quality evidence. Indeed, given the lack of prospective randomized controlled trials, which are desperately warranted, regarding the appropriateness of IVCF insertion, its role in malignancy is still a matter of debate. The purpose of this narrative review is to focus on the current literature evaluating the use of IVCF in cancer patients in an effort to provide clear insights on the insertion technique, the updated indications, and possible complications to filter placement, with particular insights on retrievable filters. This review gives emphasis to the subsets of cancer patients with advanced disease and those undergoing surgery, and it offers a “head-to-head” comparison among the main indications by major scientific societies. In summary, a conclusive, pragmatic 4-step algorithm is herein suggested to guide the treating physicians to apply the optimal IVCF management in cancer for manageable, safe, and effective decision making.

## Figures and Tables

**Figure 1 biomedicines-12-02230-f001:**
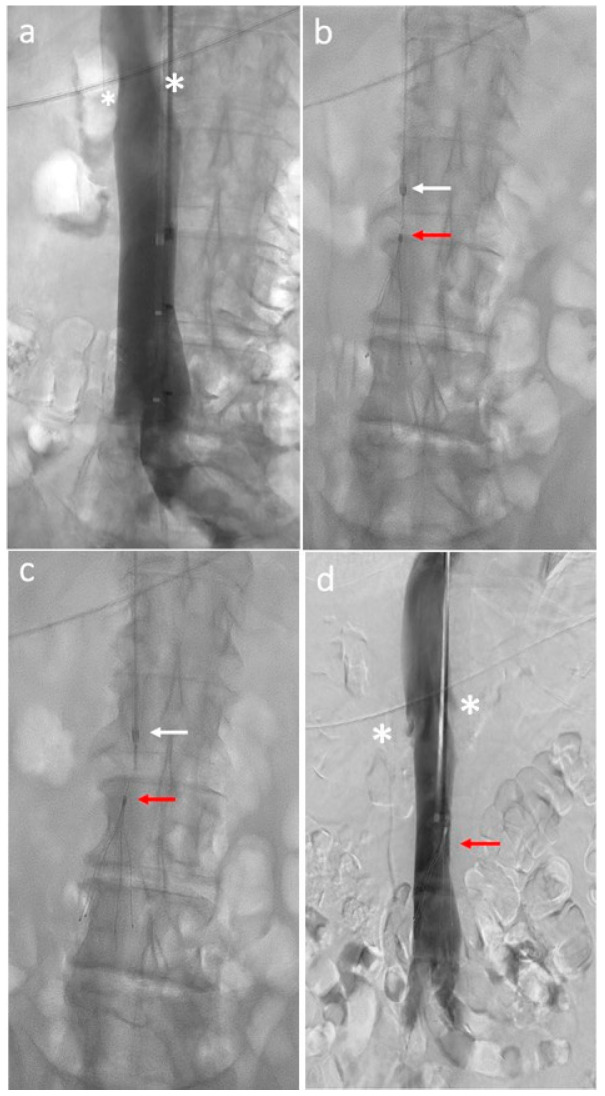
Inferior vena cava filter placement procedure: (**a**) Initial cavography shows renal veins (white asterisks) and inferior vena cava patency; (**b**) delivery sheath (white arrow) backed away several centimeters above the top of the filter (red arrow); (**c**) filter (red arrow) release from the sheath (white arrow); (**d**) final cavography shows filter (red arrow) deployment distal to the renal veins (white asterisks).

**Figure 2 biomedicines-12-02230-f002:**
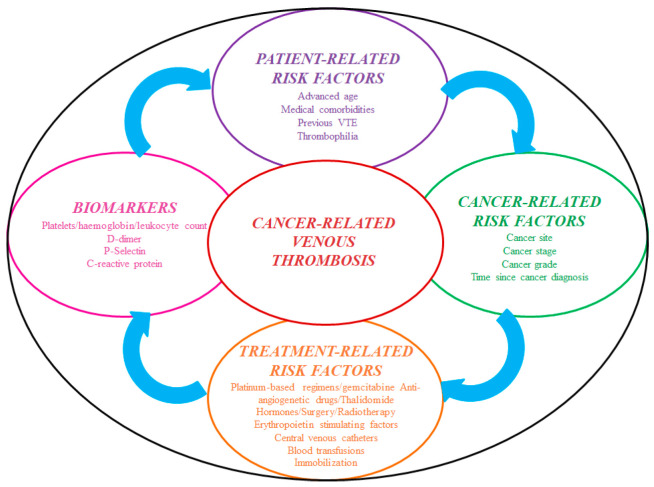
Risk factors for cancer-associated thrombosis. Abbreviations: VTE: venous thromboembolism.

**Figure 3 biomedicines-12-02230-f003:**
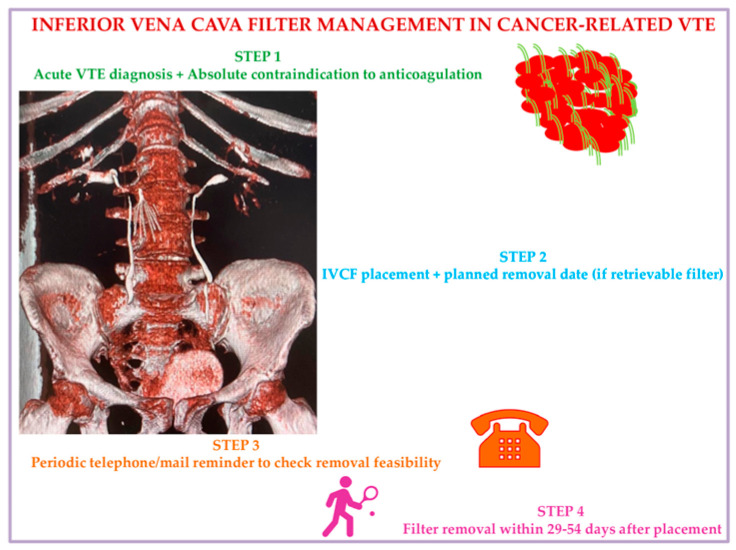
Inferior vena cava filter management in malignancy: A suggested algorithm. Abbreviations: VTE: venous thromboembolism; IVCF: inferior vena cava filter.

**Table 1 biomedicines-12-02230-t001:** Absolute, relative, and prophylactic indications to IVCF placement.

Absolute Indications in Proven VTE	Relative Indications in Proven VTE	Prophylactic Indications without VTE
Contraindications to anticoagulation	Free-floating large proximal/iliocaval DVT	High-VTE-risk surgery/invasive interventions
Anticoagulation failure	Poor compliance to manage therapeutical anticoagulation	Medical prothrombotic disease
Complications of anticoagulation	High risk to anticoagulate	High-VTE-risk trauma patients
	Massive PE/iliocaval VTE treated with thrombolysis/thrombectomy	
	VTE with limited cardiopulmonary reserve	
	Recurrent PE with IVC filter in place	
	Chronic PE treated with thromboendoarterectomy	

Abbreviations: VTE: venous thromboembolism; DVT: deep vein thrombosis; PE: pulmonary embolism; IVC: inferior vena cava.

**Table 2 biomedicines-12-02230-t002:** Inferior vena cava filter placement complications.

Short-Term Complications	Long-Term Complications
Contrast agent reaction/induced renal failure	Increased subsequent DVT risk
Arrhythmias	Filter migration/embolization/symptomatic penetration outside IVC/fracture
Air embolism	IVC stenosis/occlusion
Pneumothorax/hemothorax	Guide wire entrapment
Insertion site bleeding/hematoma/infection/thrombosis	PE/fatal PE
Arteriovenous fistula	
Filter tilting/angulation/misplacement/fracture/migration/embolization	
Guide wire entrapment/extravascular penetration	
PE/fatal PE/death	

Abbreviations: PE: pulmonary embolism; DVT: deep vein thrombosis; IVC: inferior vena cava.

**Table 3 biomedicines-12-02230-t003:** Main characteristics of studies on retrievable inferior vena cava filters.

Study	Total N. of PTS	N. of Cancer PTS	Study Design	Main Indication	Follow-Up	Results
Imberti et al. 2005 [29]	30	4	Prospective observational	Controindication to anticoagulation (86%)	18.2 months	10% trapped emboli 7% DVT recurrences
Kim et al. 2008 [31]	702	125 RF; 126 PF	Retrospective cohort study	Contraindication to anticoagulation 57.8% RF; 59.3% PF	5 years	sPE: 4.7 RF; 4% PF sDVT 12.6% RF; sDVT 11.3% PF sIVCT 0.5% RF; 1.1% PF
Buso et al. 2020 [41]	920	265	Retrospective observational study	Contraindication to anticoagulation 88.15%	12 months	mPE 0% RF; 1.2% PF Bleedings 0% RF; Bleedings4.2% PF acMortality 4.3% RF; acMortality 29.6% PF

Abbreviations: PF: permanent filter; RF: retrievable filter; sDVT: symptomatic deep vein thrombosis; sPE: symptomatic pulmonary embolism; mPE: massive pulmonary embolism; acMortality: all-cause mortality.

## Data Availability

Data sharing is not applicable.

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
