# Peer review of "Inferior Vena Cava Filter in Cancer-Associated Thrombosis: A Vade Mecum for the Treating Physicians: A Narrative Review"

_biomedicines, 2024, doi:10.3390/biomedicines12102230_

Round 1

Reviewer 1 Report

Comments and Suggestions for Authors

The authors conducted a review, summarizing the available evidence on the efficacy and safety of IVCF in cancer patients.

The topic being investigated is essential and has significant implications for clinical practice. I recommend clearly stating the type of review the authors wrote. The authors briefly classify it as a “comprehensive review” in the conclusion section, but it should also be stated in the introduction/methodology sections and the title.

The authors indirectly emphasize the need for this research by pointing out the inconsistency of information in recent years and highlighting the current research gaps and controversies surrounding using inferior vena cava filters (IVCF) in cancer patients. However, a more direct approach might be beneficial. I suggest pointing out specific areas that will be explored in the review to make the article easier to follow. Also, I recommend excluding images from the introduction and placing them in subsequent subsections instead.

I suggest adding a methodology section, pointing out the inclusion and exclusion criteria for the articles reviewed, years to search, specific populations (adults/children), included articles’ type, etc. This will increase the transparency of the research and the quality of information you provide.

The paper provides comprehensive information supported by extensive documentation. However, currently, the article dedicates a significant portion to general details on IVCFs, while only a small subsection is about their use in cancer patients. I propose devoting more space to discussing the implications for cancer patients. This will better align with the article's purpose.

The references list is exhaustive and well-chosen. I recommend adding some more up-to-date articles that have been published in the last five years. 

Author Response

Point 1

I agree with you. It is of utmost importance to state the type of review we have written as a narrative review. I have now specified it in the title/introduction/methods/conclusion chapters.

Point 2

I agree with you. At the end of the introduction chapter I clearly now listed all the sections form 3 to 9 of the review explaining the related items one-by-one, to benefit the readers with a more comprehensive analysis of the paper.

Point 3

I agree with you. I have now excluded Fig 1 from the introduction section and place it at the incipit of the section 7.

Point 4

I agree with you. I have now added section 2 “Methods” underscoring the type of the cited articles, inclusion/exclusion criteria, time and methods of search to increase the transparency and quality of the provided research that will be useful for the readers.

Point 5

I agree with you. I have now introduced implications about the use of IVCF in cancer patients in the 3 subsections of section 7 (medical cancer patients, patients with advanced cancer and surgical cancer patients) and also in chapter 8 about the usefulness of clear guidelines for a more accurate approach to IVCF use. I have also added a hallmark study of cancer associated thrombosis (Prandoni 2002) to focus on the implications of IVCF in cancer patients with bleeding considerations.

Point 6

I agree with you. I replaced reference 1 Khorana 2010 with Khorana 2022. I have now introduced 7 more up-to-date articles (Pandhi 2016, Kurzyna 2024, Abdel-Razeq 2023 and Makary 2024 in section 7, Gabara 2023 and Bajda 2023 in section 8, Benedetti 2024 in section 9)

Reviewer 2 Report

Comments and Suggestions for Authors

The review article “ Inferior vena cava filter in cancer associated thrombosis: a vade mecum for the treating physicians. A review” is interesting, as the information provided can be very useful for the readers. I have the following comments/suggestions,

1.      Define VTE is the abstract.

2.      It will be more appropriate to include the type of review in the title or in the abstract.

3.      In the last paragraph of the introduction section, it will be more interesting for the readers to know how the information was collected for the performed review.

4.      Table 1, define the abbreviations in the footnote.

5.      Figure 2 can be made more interesting by joining the gap between a,c & b,d. Moreover, the part b and d and not aligned.

6.      Table 2, define the abbreviations.

7.      5. Retrievable vena cava filters, & 6.1 Evidence among medical cancer patients, these parts are too lengthy the authors are advised to make 2 or 3 paragraphs out of it as it is very hard to follow.

Author Response

Point 1

I agree with you. I have now defined VTE in the abstract chapter

Point 2

I agree with you. I have now specified it as a narrative review article in the title/introduction/methods/conclusion chapters.

Point 3

I agree with you. I have now added section 2 “Methods” underscoring the type of included articles, inclusion/exclusion criteria, time and methods of search to increase the transparency and quality of the provided research that will be useful for the readers.

Point 4

I agree with you. I have added footnotes at the bottom of Table 1 and I defined the abbreviations.

Point 5

I agree with you. I tried to put in a unique line the 4 pictures of Figure 2 joining the gaps but the resolution of the images was markedly reduced and I decided to divide the 4 pictures into 2 groups (group 1: a and b; group 2: c and d) to enable the readers enjoying higher quality and more clear images. I have also now aligned picture b with picture d.

Point 6

I agree with you. I have now added footnotes at the bottom of Table 2 and I defined the abbreviations.

Point 7

I agree with you. I have now erased the details on the different technical features among the several types of retrievable filters (section 6) to streamline the text. We wanted to deeply mention all the most recent articles on the retrievable filters because they are now emergent in clinical practice and, although these studies have been conducted in the general population, we underscored one-by-one the percentage of cancer patients included. I have now also cancelled some sentences from section 7 subsection 1 on the medical cancer patients, although all the articles cited in this section are very precious and useful to be known by all the treating physicians since literature is generally very scant on IVCF in malignancy.

Reviewer 3 Report

Comments and Suggestions for Authors

In this paper, the authors review the state of the art on using inferior vena cava filter (IVCF) in cancer patients. Different aspects of IVCF including indications to filter placement, placement procedure, complications, safety, and outcomes on cancer patients are covered. They also discuss the guidelines and management related to IVCF. Moreover, they provide a 4-step practical strategic algorithm to guide the treating physicians in applying the optimal IVCF on cancer patients. Overall, the review is comprehensive on the topic. Regarding the organization, the different case studies in section 5 (retrievable vena cava filters) should be tabulated for better interpretation. For the references, several most recent related research articles and review papers need to be added:

1. Semin Intervent Radiol 2016;33:71–74. DOI http://dx.doi.org/10.1055/s-0036-1581090

2. Cancers 2024, 16, 1562. https://doi.org/10.3390/cancers16081562

3. Bajda J, Park A N, Raj A, et al. (June 06, 2023) Inferior Vena Cava Filters and Complications: A Systematic Review. Cureus 15(6): e40038. DOI 10.7759/cureus.40038

4. Angiology. 2023;0(0). doi:10.1177/00033197231190184

5. J. Clin. Med. 2024, 13, 1761. https://doi.org/10.3390/jcm13061761

6.  J Surg Oncol. 2024; 130: 257-264. doi:10.1002/jso.27734

7. J. Clin. Med. 2023, 12, 7209. https://doi.org/10.3390/jcm12237209

Author Response

Point 1

I agree with you. I tabulated the articles on the retrievable filters in section 6 in Table 3; however, most of them are reviews, registries, audit that don’t fit very well in a tabled list.

Point 2

I agree with you. I introduced in the text and in the bibliography the 7 up-to-date articles you mentioned and I really thank you to share them with us because I found them very useful to enrich our manuscript.